# Ranking-Aware Multiple Instance Learning for Histopathology Slide Classification

**Ho Heon Kim**[*1] [ID]                                     HOHEON0509@MF.SEEGENE.COM

**Gisu Hwang**[*1] [ID]                                          GSHWANG@MF.SEEGENE.COM

**Won Chang Jeong**[1] [ID]                          JEONGWONCHAN53@MF.SEEGENE.COM

**Young Sin Ko**[†1,2] [ID]                                     NOTEASY@MF.SEEGENE.COM

[1] *AI Research Center, Seegene Medical Foundation, 320 Cheonho-daero, Seoul, South Korea*

[2] *Pathology Center, Seegene Medical Foundation, 320 Cheonho-daero, Seoul, South Korea*

**Editors:** Accepted for publication at MIDL 2025

## Abstract

In digital pathology, most deep learning models adopt multiple instance learning (MIL) as it requires only slide-level labels, reducing the need for detailed annotations. However, since MIL still relies on large datasets, data-efficient strategies have emerged as promising alternatives. Although some datasets include expert annotations, their integration with MIL to take advantage of this valuable information has been overlooked. We propose `Rank induction`[1], a method that ranks annotated lesion areas against non-lesion areas to guide the model's attention toward diagnostically meaningful areas. Our experiments on the Camelyon16 dataset show that `Rank induction` outperforms existing approaches in classification performance. Furthermore, the method remains robust under data-scarce conditions. Finally, attention maps generated by the model trained with `Rank induction` focus more accurately on cancerous areas.

**Keywords:** Multiple instance learning, learning to rank, digital pathology.

## 1. Introduction

With the growing adoption of digital pathology, deep learning has demonstrated clinical benefits across tasks ranging from tumor detection to genotype prediction. However, analyzing whole slide images (WSIs) remains challenging due to their gigapixel scale and the sparsity of lesion areas, making manual annotation costly and time-consuming (Gadermayr and Tschuchnig, 2024). To address this, many approaches rely on multiple instance learning (MIL), which requires only slide-level labels. Yet, MIL still depends on large-scale WSIs, and while data-efficient methods such as pseudo-labeling have been explored.

Despite recent progress, most MIL frameworks rarely incorporate expert annotation, even when such annotations are available and can provide stronger supervision. One notable exception is attention induction (Koga et al., 2025), which makes a strong assumption that attention weights should exactly match the proportion of lesion areas. However, this assumption may not hold in histopathology, where lesion size does not necessarily reflect diagnostic importance. To address this, we propose `Rank induction`, a weaker and more pathology-informed assumption. Our method guides the model to assign higher attention to annotated lesion patches than to non-lesion ones by enforcing a ranking constraint, leveraging expert annotations.

---

[*] Contributed equally

[†] Corresponding author

1. The code is available at: https://github.com/SMF-AI/rank_induction

## 2. Methods

### 2.1. Problem Definition

In contrast to the *standard MIL* assumption—which assumes latent instance labels and deems a bag positive if it contains at least one positive instance—our setting provides explicit instance-level annotations for instance-level supervision (Maron and Lozano-Pérez, 1998). A WSI is represented as a set of patch images $\mathcal{X} = \{x_k\}_{k=1}^{K}$, associated with a slide-level label $Y \in \{0, 1\}$. Instance-level labels are denoted as $\mathcal{Y} = \{y_k\}_{k=1}^{K}$, where $y_k \in \{0, 1\}$ indicates whether the patch $x_k$ belongs to the lesion areas manually annotated by a pathologist.

### 2.2. Proposed Method

Inspired by ranknet (Burges, 2010), `Rank induction` guides the model by imposing a ranking constraint, encouraging higher attention scores for annotated lesion patches compared to non-lesion ones: $s_i > s_j \quad \forall i, j$ such that $y_i = 1$, $y_j = 0$. To implement this, we compare raw attention scores $s_i \in \mathbb{R}$ before softmax normalization, as normalized attention weights $a_i \in [0, 1]$ are constrained by $\sum_k a_k = 1$, making it difficult to enforce a large margin. The score difference is mapped to a pairwise probability via:

$$P_{i,j} \equiv \frac{1}{1 + e^{-\sigma(s_i - s_j - m)}}$$

where $m$ is a margin parameter that encourages a greater separation between lesion and non-lesion patches in terms of attention scores, and $\sigma$ is a scaling factor. Considering only lesion–non-lesion comparisons and excluding same-class pairs (i.e., lesion–lesion or non-lesion–non-lesion), we define the valid index pairs $\mathcal{P} = \{(i, j) \mid y_i = 1, \ y_j = 0\}$. Let $\bar{P}_{i,j} \in \{0, 1\}$ denote the ground truth ranking preference. The rank loss is defined as:

$$\mathcal{L}_{\text{rank}} = \frac{1}{|\mathcal{P}|} \sum_{(i,j) \in \mathcal{P}} -\bar{P}_{i,j} \log P_{i,j} - (1 - \bar{P}_{i,j}) \log(1 - P_{i,j})$$

We formulate the objective function by jointly optimizing the slide-level classification loss (BCE) and the ranking loss. To prevent shortcut learning—where the model might trivially prioritize lesion patches while completely disregarding the signals of non-lesion patches—we apply attention thresholding to encourage the model to retain information from both lesion and non-lesion areas: $a_k := max(a_k - T/K, 0)$, where $T$ is a threshold. For memory-efficient training, pairs $(i, j)$ were sampled from $\mathcal{P}$.

### 2.3. Data and experiments

We conducted experiments on the Camelyon16 (Bejnordi et al., 2017) dataset for breast lymph node metastasis detection, which consists of 399 WSIs—270 for training and 129 for testing. Each WSI was tessellated at $20\times$ magnification into patches of size $224\times224$. We used a ResNet-50 model pretrained on ImageNet, applying adaptive average spatial pooling after the third residual block. The proposed model was evaluated under two settings: (1) improvement in slide-level classification performance, and (2) the effectiveness of instance-level supervision under limited data scenarios by sampling the training set at specific ratios,

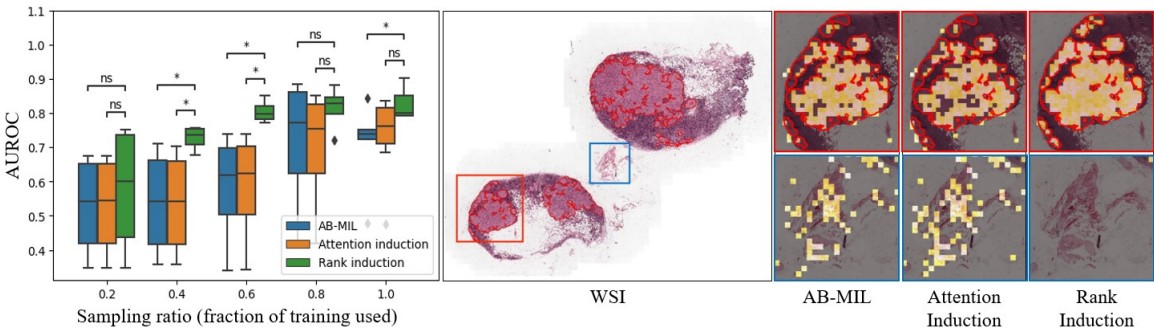

Figure 1: Model performance under data scarcity and interpretation. Left: Performance at varying sampling ratios. Middle: WSI thumbnail. Right: Heatmap. The red boxes highlight cancerous areas; the blue boxes indicate non-lesion areas.

reflecting real-world conditions where data scarcity is common and expert annotations can be leveraged to compensate for small datasets. To reduce the effect of randomness, experiments were repeated 10 times (Monte Carlo cross-validation) for the full dataset and 5 times for data scarcity scenarios. The Mann–Whitney U test was used for statistical comparison.

## 3. Results

The model training with `Rank induction` showed 0.836 of AUROC and 0.851 of AURPC, which statistically significantly outperformed AB-MIL (Ilse et al., 2018)(without annotation) and attention induction ($p < 0.05$, respectively, Table 1) with low variance. Notably, this performance was consistently replicated on the DigestPath dataset (Table S1).

Table 1: Comparison of model performance across methods

| Metric | Methods | | | P value | |
|---|---|---|---|---|---|
| | AB-MIL | Attention induction | Rank induction | AB-MIL *vs* Rank induction | Attention induction *vs* Rank induction |
| AUROC | 0.740 (± 0.146) | 0.743 (± 0.142) | 0.836 (± 0.044) | 0.032 | 0.032 |
| AUPRC | 0.730 (± 0.183) | 0.727 (± 0.179) | 0.851 (± 0.036) | 0.027 | 0.012 |
| Accuracy | 0.778 (± 0.093) | 0.770 (± 0.080) | 0.842 (± 0.012) | 0.037 | 0.017 |

Figure 1 shows that our approach maintains stable performance across varying sampling ratios (left) and produces more accurate, interpretable attention maps (right). Additionally, `Rank induction` demonstrated robustness to coarse annotations (Figure S1).

## 4. Conclusion & Future work

In this work, we proposed `Rank induction`, a ranking-based attention supervision method for MIL in digital pathology. By leveraging expert annotations through pairwise ranking, our approach improves performance and interpretability, even in data-limited scenario. In future work, we plan to investigated its generability across multiple datasets, comparison it with more existing methods.

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

## Appendix A. Model comparison in DigestPath dataset

As an additional experiment, we evaluated `Rank induction` on the DigestPath dataset, which comprises 660 colonoscopy histopathology images—250 tumor-positive and 410 tumor-negative. Each image has an average resolution of approximately 5000×5000 pixels. We cropped each image into non-overlapping 224×224 patches and discarded background regions to retain only tissue areas. To ensure robustness and mitigate sampling bias, we employed stratified Monte Carlo cross-validation, repeating the evaluation with ten different random seeds. In each split, we maintained an 80:20 ratio between the training and test sets while preserving the original class distribution.

`Rank induction` achieved an AUROC of 0.995 (+0.003, +0.002, compared to AB-MIL, attention induction respectively), which, although not always statistically significant, consistently yielded the highest performance among the compared methods (Table S1). For AUPRC, `Rank induction` reached 0.993 (+0.004, +0.003 compared to the two other methods), significantly outperforming the alternatives. Despite the relatively small image size in the DigestPath dataset, which makes the task less challenging and causes performance saturation across models, `Rank induction` still demonstrated the best overall results.

Table S1: Comparison of model performance across methods on DigestPath dataset

| Metric | Methods | | | *P* value | |
|---|---|---|---|---|---|
| | AB-MIL | Attention induction | Rank induction | AB-MIL *vs* Rank induction | Attention induction *vs* Rank induction |
| AUROC | 0.992 (± 0.002) | 0.993 (± 0.002) | 0.995 (± 0.001) | 0.034 | 0.060 |
| AUPRC | 0.989 (± 0.004) | 0.990 (± 0.003) | 0.993 (± 0.002) | 0.037 | 0.027 |
| Accuracy | 0.963 (± 0.017) | 0.965 (± 0.012) | 0.967 (± 0.013) | 0.336 | 0.335 |

## Appendix B. Comparison of Annotation Granularity

In an additional experiment to investigate the association between annotation granularity and model performance, we generated synthetic expert annotations by expanding existing polygon-based annotations. During this process, each polygon was padded outward in all directions in multiples of 448 pixels at $40\times$ magnification, resulting in the inclusion of additional noise patches along the top, bottom, left, and right edges. This corresponds to adding one noise patch per edge at $20\times$ magnification. An example of this padding is illustrated at the bottom of (Figure S1). Despite the presence of such coarse (and potentially inaccurate) annotations, `Rank induction` showed minimal performance degradation, demonstrating its robustness to annotation imprecision (Figure S1).

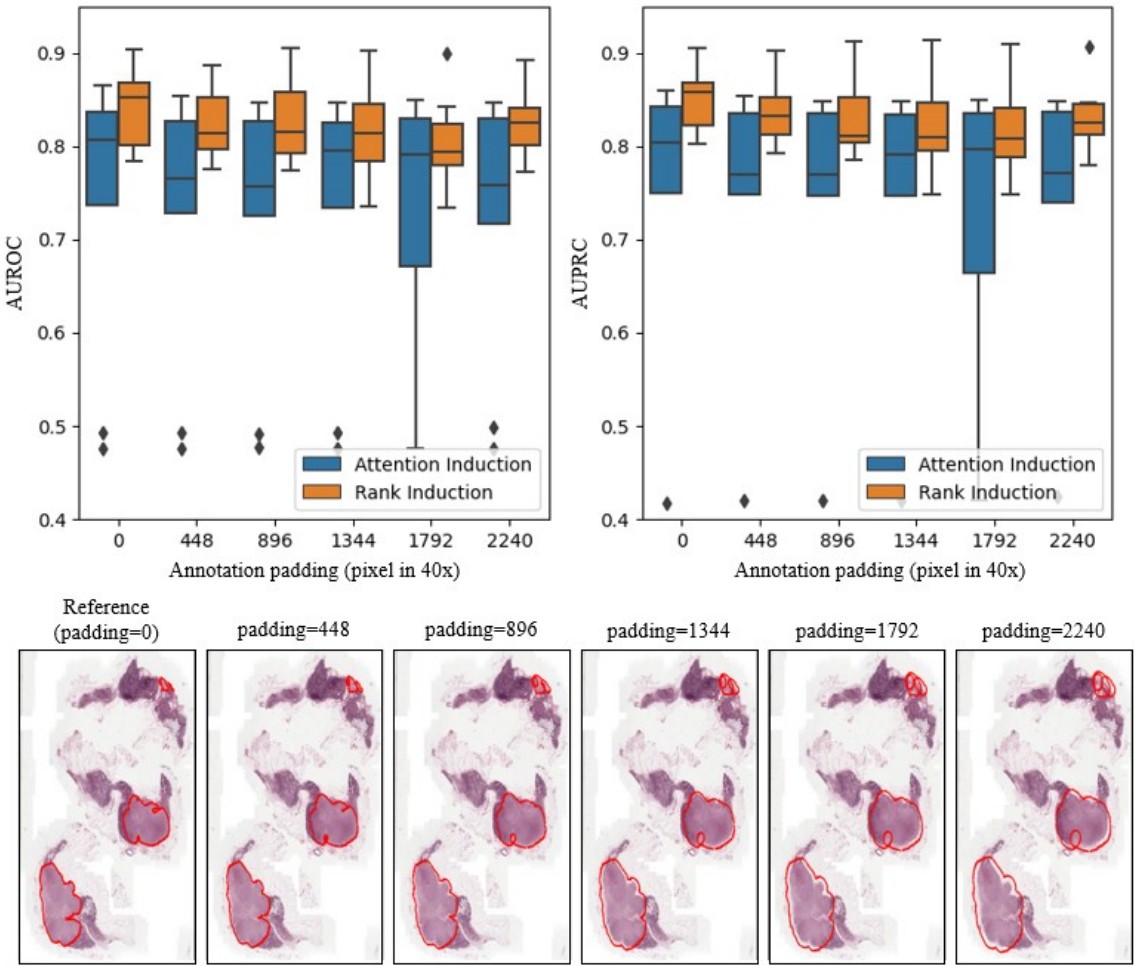

Figure S1: Comparison of model performance by annotation granularity. Top-left: AUROC of Attention Induction and Rank Induction. Top-right: AUPRC of Attention Induction and Rank Induction. Bottom: Example of annotation granularity by padding size

