# OpenReview forum: "Ranking-Aware Multiple Instance Learning for Histopathology Slide Classification"
_MIDL.io/2025/Short_Papers — MIDL 2025 - Short Papers_

### Official Review · Reviewer_nHhY · 2025-04-28

**Rating:** 5
**Confidence:** 4

**Summary:**

This paper introduces Rank Induction for data-efficient multiple instance learning on whole slide images (WSIs). The authors collect instance-level labels to guide attention mechanisms toward lesion areas, aiming to improve classification performance. Extensive experiments are conducted to validate the proposed approach.

**Strengths:**

•	The paper is well-organized, and the methodology is clearly presented.
•	Rather than simply adopting existing models, the authors propose meaningful modifications.
•	The experimental results are thorough, and the release of code adds to the paper’s reproducibility.

**Weaknesses:**

•	Since multiple instance learning typically aims to avoid the need for instance-level annotations, relying on such annotations reduces the practicality of the proposed method.
•	Given the availability of instance-level labels, comparisons with classical patch-based classification approaches would strengthen the evaluation.

---

### Decision · Program_Chairs · 2025-05-01

Accept